# BMN673 Is a PARP Inhibitor with Unique Radiosensitizing Properties: Mechanisms and Potential in Radiation Therapy

**DOI:** 10.3390/cancers14225619

**Published:** 2022-11-16

**Authors:** Aashish Soni, Xixi Lin, Emil Mladenov, Veronika Mladenova, Martin Stuschke, George Iliakis

**Affiliations:** 1Division of Experimental Radiation Biology, Department of Radiation Therapy, University Hospital Essen, University of Duisburg-Essen, 45147 Essen, Germany; 2Institute of Medical Radiation Biology, University Hospital Essen, University of Duisburg-Essen, 45147 Essen, Germany; 3German Cancer Consortium (DKTK), Partner Site University Hospital Essen, German Cancer Research Center (DKFZ), 45147 Essen, Germany

**Keywords:** PARP inhibitor, BMN673, talazoparib, olaparib, radiotherapy, radiosensitization, DNA double strand breaks (DSB), DSB repair, ionizing radiation, c-NHEJ, alt-EJ, homologous recombination, DNA end-resection, error-prone DSB processing

## Abstract

**Simple Summary:**

PARP inhibitors (PARPi) have broad applicability as monotherapy in tumors with HR defects. However, their combination with radiotherapy (RT) is less advanced owing to the rather modest radiosensitization typically achieved with most tested PARPi, including olaparib. Tumor resistance to PARPi or RT is a significant bottleneck in the clinic, and strategically designed combinations of PARPi and RT may offer approaches to overcome such resistance. Talazoparib (BMN673), as compared to other PARPi, is a superior radiosensitizer. In this review, we discuss our evolving understanding of the mechanistic underpinnings of BMN673 radiosensitization and outline the potential for combinations of BMN673 with RT to treat various forms of cancer.

**Abstract:**

BMN673 is a relatively new PARP inhibitor (PARPi) that exhibits superior efficacy in vitro compared to olaparib and other clinically relevant PARPi. BMN673, similar to most clinical PARPi, inhibits the catalytic activities of PARP-1 and PARP-2 and shows impressive anticancer potential as monotherapy in several pre-clinical and clinical studies. Tumor resistance to PARPi poses a significant challenge in the clinic. Thus, combining PARPi with other treatment modalities, such as radiotherapy (RT), is being actively pursued to overcome such resistance. However, the modest to intermediate radiosensitization exerted by olaparib, rucaparib, and veliparib, limits the rationale and the scope of such combinations. The recently reported strong radiosensitizing potential of BMN673 forecasts a paradigm shift on this front. Evidence accumulates that BMN673 may radiosensitize via unique mechanisms causing profound shifts in the balance among DNA double-strand break (DSB) repair pathways. According to one of the emerging models, BMN673 strongly inhibits classical non-homologous end-joining (c-NHEJ) and increases reciprocally and profoundly DSB end-resection, enhancing error-prone DSB processing that robustly potentiates cell killing. In this review, we outline and summarize the work that helped to formulate this model of BMN673 action on DSB repair, analyze the causes of radiosensitization and discuss its potential as a radiosensitizer in the clinic. Finally, we highlight strategies for combining BMN673 with other inhibitors of DNA damage response for further improvements.

## 1. Introduction

Radiotherapy (RT) and chemotherapy are two main pillars of cancer treatment. As the primary treatment modality, RT contributes to about 40% of curative treatments for cancer [1,2]. Therapeutically exploiting DNA repair offers a potent means to target cancer cells [3,4]. PARP inhibitors (PARPi) are widely used chemotherapeutic agents in homologous recombination (HR) deficient tumors. The seminal finding of a synthetic lethal interaction between PARP-1 and HR defects [5,6] accelerated the development of highly specific, clinically relevant PARPi, including olaparib (AZD-2281), niraparib (MK-4827), veliparib (ABT888), rucaparib (AG-014699), and talazoparib (BMN673) [7,8,9]. Such clinical PARPi exclusively target PARP-1 and PARP-2 to execute their anti-cancer effects through a synthetically lethal interaction with HR defects. The idea of synthetic lethal interactions between the PARPi and HR defects has been exploited as an Achilles heel in the management of ovarian and breast cancers and has been extensively reviewed in the past [10,11,12,13,14,15,16,17].

The PARP family of proteins regulates several biochemical processes in a living cell by mediating a post-translational modification called poly- or mono-ADP-ribosylation (PARylation or MARylation). Where PARP-1, 2, 5a, and 5b are capable of modifying their targets through PARylation, the remaining PARP family members modify their targets through MARylation [18]. Among 17 reported members of the PARP family [19], only PARP-1, PARP-2, and PARP-3 were found to regulate DNA damage response and repair (Figure 1). However, the main target of PARPi is PARP-1, which regulates DSB repair through PARylation of various signaling proteins, including itself. PARP-1 is the critical protein involved in single-strand break repair (SSB) and alternative end joining (alt-EJ) [20]. Multiple studies extend the primary functions of PARP-1 in DNA repair and suggest a broader role for PARP-1 and PARylation in the complex network of multiple DNA repair mechanisms [21,22]. This has highlighted PARP-1 as a central molecule coordinating the overall cellular responses to genotoxic stress in general and DNA damage, in particular (DDR) [23].

Tumors lacking HR and therefore therapeutically treated with PARPi, frequently acquire resistance when treated for a prolonged period with such inhibitors [24]. The prime mechanisms behind development of PARPi resistance in tumors are restoring HR activity, losing PARP-1 activity, or upregulating alt-EJ repair functions [24]. Combining PARPi with other chemotherapeutics or RT may help in preventing or overcoming PARPi resistance, increasing thus the tumor response to therapy. 

Combination with RT offers an attractive scenario for enhancing the efficacy of PARPi [25,26,27,28,29]. Several studies have shown potential for radiosensitization for several clinically relevant PARPi, however, most of them show only moderate radiosensitization [30,31,32,33,34,35,36,37,38] except for BMN673. BMN673 showed robust radiosensitization in cancer cells [39,40,41], while leaving normal cells almost unaffected [39]. BMN673 exerts distinct DSB repair modulatory activities and appears to offer surprisingly flexible drug treatment schedules that make it the “ideal” clinical radiosensitizer. The expansion of the clinical use of PARPi in the face of acquired resistance, rationalizes the active search for clinical strategies combining PARPi with other agents able to suppress, mitigate or delay resistance development. RT is a promising candidate along these lines. Here, we discuss the rationale and potential of combining PARPi and RT with particular emphasis on the ability of BMN673 to strongly radiosensitize tumor cells and xenografts, paving the way for clinical and pre-clinical tests. 

## 2. PARP-1 Operates at the Heart of DNA Damage Response and DNA Repair

Exacerbation occasional deregulation, or suppression of the DNA repair machinery is a promising avenue for improvements in cancer treatment utilizing agents generating damage in the DNA. The genomic integrity of higher eukaryotes is continuously challenged by chemical and physical agents capable of introducing covalent DNA modifications [20,42]. Mammalian cells have evolved well-coordinated DNA damage response (DDR) signaling to protect genomic DNA [43,44,45,46,47,48]. DDR is particularly sensitive to DSBs and replication stress and orchestrates the engagement of DSB repair pathways, while coordinating at the same time their repair with cell cycle checkpoints. It is well-documented that PARP-1 is tightly associated with chromatin. PARP-1 is endowed with high detection sensitivity to DNA damage, particularly single strand breaks (SSBs) and DSBs, causing its rapid activation that generates large amounts of PAR-chains on several proteins, including itself [19,49,50]. PARP-1-mediated PARylation ensures the timely recruitment of DNA repair proteins to DSBs [22,51]. PARP-1 plays a crucial role as a DNA damage sensor in DDR activating crucial enzymes, operating as transducers and effectors of DDR signaling (Figure 2) [52]. 

In genetic studies, PARP-1 was originally identified as a key protein in the base excision repair (BER) pathway [53,54,55,56,57,58]. BER removes from DNA oxidative base damage, alkylation damage and abasic sites through the synchronized actions of DNA glycosylases, apurinic-apyrimidinic (AP) endonucleases, end-processing enzymes (polynucleotide kinase-phosphatase (PNKP) and flap endonuclease (FEN-1)), DNA polymerases and the DNA Ligase3/XRCC1 complex [59,60,61,62]. PARP-1 inhibition suppresses the oxidative stress response in human lens cells by inhibiting BER [63]. Several studies, on the other hand, suggest that PARP-1 is not essential for BER [22,54,64,65].

PARP-1 promptly detects single-strand breaks (SSBs) through direct interaction with the damaged DNA [22,66,67]. One of the crucial roles of PARP-1 in SSB repair is to recruit the Ligase 3/XRCC1 complex, which mediates DNA ligation. It has been demonstrated that the recruitment of XRCC1 to SSBs is PARP-1 and PARP-2 dependent [68,69] and that mutation in XRCC1 results in PARP-1 hyperactivity, which leads to exhaustion of the cellular NAD^+^, causing cell death. PARP-1 can also stimulate the gap-filling step and the final DNA ligation step of SSB repair, by intervening with the availability of ATP [70]. 

Recent studies have also suggested a role for PARP-1 in DSB repair pathways: classical non-homologous end joining (c-NHEJ), homologous recombination (HR), alternative end joining (alt-EJ), and single-strand annealing (SSA). The PARP-1 requirements for c-NHEJ function are less well characterized. It has been demonstrated that PARP-1 can PARylate the DNA-dependent protein kinase catalytic subunit (DNA-PKcs), which is the primary coordinator of c-NHEJ [71]. This study shows that PARylation of DNA-PKcs stimulates its kinase activity without requiring interaction with the KU70/80 heterodimer [71]. Moreover, it has been demonstrated that PARP-1 competes with the KU70/80 heterodimer on DSBs, thus negatively regulating c-NHEJ [72]. This suggests that PARP-1 may even shift DNA repair towards DNA end-resection-dependent pathways. 

The role of PARP-1 in HR is somewhat complex, and earlier studies showed that PARP-1 is dispensable for HR. In PARP-depleted cells, HR was normal, and PARylation showed a little direct effect on HR [73]. Inhibition or depletion of PARP-1 resulted in spontaneous sister chromatid exchanges (SCEs). As SCEs are strongly dependent on HR, this result suggests that PARP-1 inhibits HR [74]. On the other hand, PARP-1 depletion failed to suppress gene conversion in an I-SceI-induced DSB in the specifically designed DR-GFP reporter cell line [39,73]. There is evidence that PARP-1 is required to efficiently detect DSBs, by interacting with MRE11 and the apical DDR kinase ATM [22,51]. These interactions implicate PARP-1 in DNA end-resection, the initial step of DSB-repair by HR. It was also shown that PARP-1 recruits MRE11 and RAD51, which restart stalled replication in a BRCA1/2-dependent manner [75]. The accumulation of replication protein A (RPA) at the generated single-stranded DNA tracts during DNA end-resection requires the activity of BRCA1. There is evidence that PARP-1 acts as a resection stimulatory factor also by interacting with BARD1, an auxiliary factor of BRCA1 that helps recruit BRCA1 to DSBs. PARP-1 activity is also detected at later HR steps, where the PARylated BRCA1 limits DNA end-resection by recruiting the RAP80 chromatin factor. The interaction with RAP80 stabilizes BRCA1 and suppresses HR, resulting in effective strand invasion and subsequent break repair [76]. Collectively, these observations suggest stimulatory and repressive functions for PARP1 in DNA end-resection. 

PARP-1 has a well-established role in alt-EJ. Abrogation of PARP-1 activity by chemical inhibition or transcriptional inactivation reduces the frequency of alt-EJ-mediated chromosomal translocations, suggesting that PARP-1 is an alt-EJ factor [39,77,78,79]. However, in PARP-1 deficient genetic background, radiosensitization and DSB repair inhibition at high IR doses are robust mainly in Ku deficient cells [72]. This notion is supported by the avid PARP-1 recruitment at DSBs in KU deficient cells and the increased loading of MRN and XRCC1/LIG3 complex at DSBs [80]. In addition, there are reports attributing a role to PARP-1 in the recruitment of DNA polymerase θ (Pol θ), a molecule recently identified as a component of alt-EJ [81,82]. Advances in crystallography and the ongoing characterization of the functional network of PARP-1 interacting proteins, open new avenues for developing specific inhibitors to target PARP-1 activity, or to disrupt interactions with protein factors of alt-EJ.

Although the function of PARP-1 in SSA is scarcely documented, studies indicate a minor, indirect role for PARP-1 in this process. For example, it has been reported that deficiency of ERCC1, a component of the excision nuclease complex (ERCC1-XPF), activated during SSA, results in synthetic lethality after olaparib treatment [83]. Moreover, reports suggest that PARP-1 modulates the activity of FEN-1; a protein also involved in RAD52-mediated SSA [84,85]. However, as PARP-1, ERCC1-XPF and RAD52 are shown to be involved in multiple DNA repair pathways, this makes it difficult to establish specific roles of PARP-1 in SSA.

## 3. Development of Clinically Relevant PARP Inhibitors

The discovery of synthetically lethal interactions between BRCA1/BRCA2 loss of function mutations and PARP inhibition paved the way to the development as a monotherapy of clinically relevant PAPRi with strong and specific activity. Olaparib (Lynparza^®^, developed by AstraZeneca, Cambridge, UK) was the first PARPi approved in 2014 by the European Medicines Agency (EMA) and by the Food and Drug Administration (FDA) in the United States for use as a single agent in the clinic. It showed a favorable response in patients with germline BRCA1/2-mutated advanced, recurrent ovarian cancer, and for maintenance treatment of adult patients with recurrent gynecologic cancers [86,87,88,89,90]. To date, three additional PARPi have been approved by the FDA. Rucaparib (Rubraca^®^, developed by Clovis Oncology, Boulder, CO, USA) was approved in 2016 for the treatment of ovarian cancer in patients with somatic and germline BRCA mutations, and in maintenance treatment settings [91,92]. Niraparib (Zejula™, developed by Tesaro, Waltham, MA, USA) was approved in 2017 for maintenance therapy of adult patients with recurrent gynecologic cancers, irrespective of BRCA mutations, or HR status [93]. Talazoparib (BMN673, Talzenna, developed by Pfizer, New York, NY, USA) was approved by the FDA (2018) and EMA (2019) for the treatment of germline BRCA-mutated, HER2-negative metastatic breast cancer [9,94]. Among all these clinical PARPi, BMN673 has shown superior efficacy in vitro [95] and may offer a significant clinical benefit as monotherapy or in combination with other anticancer agents.

## 4. Rationale for Combining PARP Inhibitors with Radiotherapy

RT has been one of the prime cancer treatment modalities [1,2]. While tumors occasionally develop radioresistance, compromising treatment efficacy, the likelihood of this development is lower than after treatment with chemotherapeutics. Altered DNA repair activity is the primary radioresistance mechanism, and targeting repair mechanisms with chemical inhibitors offers therefore attractive treatment possibilities. Such inhibitors may target specific pathways utilized by irradiated cells to repair DSBs. As described above, PARP-1 is involved in multiple DNA repair pathways, including the repair of SSBs and DSBs (Figure 2). Therefore, its inhibition is expected to cause radiosensitization. It is also relevant that elevated expression of PARP-1 is reported in several radioresistant tumors [35,96,97]. Combining, therefore, PARPi with RT may offer means to radiosensitize cells or to overcome radioresistance when it develops. 

When PARPi are used as a monotherapy in HR deficient tumors for an extended period, resistance to PARPi develops by reduced PARP-1 levels [98,99], restoration of HR, or the acquisition of compensating mutations [100,101,102,103]. The overall tumor response rate to PARPi ranges from 30 to 50% [104], indicating that tumors either possess de novo resistance, or develop resistance later. Combining PARPi only during the course of RT may prevent the development of resistance owing to the short course of application (usually about 8 weeks). Additionally, PARPi may radiosensitize tumors with functional HR offering thus means to treat tumors irrespective of their HR status. Indeed, several pre-clinical and clinical studies have tested the efficacy of various PARPi-RT combinations [26]. Recently, radiolabeled PARP inhibitors have been developed and tested for targeted tumor therapy [105]. In a pre-clinical study, encouraging results have been recently reported in EGFR-mutated NSCLC, when niraparib was combined with radiation [106]. 

## 5. BMN673 Is a Superior Radiosensitizer

A few pre-clinical studies have shown the potential of BMN673 in combination with RT on multiple cancer cell lines and xenografts. BMN673 was shown to inhibit the proliferation of glioblastoma stem cells after exposure to low or high linear energy transfer radiations [107]. In 2018, we were the first to report the strong radiosensitizing potential of BMN673 (50 nM) in a battery of cancer cell lines originating from Rhabdoid tumors, Sarcomas, lung carcinomas, osteosarcomas, and colon cancer [39]. We found that only a short drug treatment time (~1 h) prior to RT was sufficient to achieve robust radiosensitization. Under similar experimental conditions, olaparib (3 µM), AG14361 (0.4 µM), and PJ34 (5 µM) showed only moderate radiosensitization. A potent radiosensitizing effect of BMN673 (200 nM) but not veliparib (1600 nM) was also reported in small cell lung carcinoma (SCLC) cell lines and Xenografts [40]. The non-small cell lung carcinoma (NSCLS) cell line, H460, was strongly radiosensitized after treatment with 2 nM BMN673, which could be further enhanced by treating cells with 5-Azacytidine [108]. Colorectal cancer cells and xenografts (BRAF wild type and mutant) could also be strongly radiosensitized by BMN673 (125 nM), whereas olaparib (250nM) and rucaparib (250 nM) exerted only a moderate radiosensitizing effect [109]. Highly radio- and chemotherapy-resistant chondrosarcoma cells were also shown to be strongly radiosensitized by BMN673 (5 nM) irrespective of their IDH (isocitrate dehydrogenase) mutation status [41]. Hepatocellular carcinoma cells, HepG2 and PLC/PRF/5, were also robustly radiosensitized with BMN673 (50 nM), whereas veliparib (10 µM) was ineffective [110]. A battery of melanoma cell lines was also strongly radiosensitized by BMN673 (50 nM). In the follow-up study, niraparib could not reach similar levels of radiosensitization, even at a concentration of 2500 nM [111].

One of the prominent features of BMN673, evident in the above-mentioned studies, is that it can be used at low nanomolar concentrations to achieve strong radiosensitization. Such concentrations are achievable in the blood of treated patients also ensuring in vivo radiosensitization [112]. No other clinical PARPi showed such efficacy and similar pharmacological properties, as pointed out earlier in this section. Not surprisingly, therefore, in pre-clinical and clinical settings, BMN673 showed robust antitumor activities at relatively low concentrations. For instance, olaparib was administered at 50 mg/kg, whereas BMN673 was used at a much lower dose of 0.3 mg/kg in pre-clinical studies (8). Similar trends were also noted in the clinical setting (protocols: NCT02032823, NCT02184195, NCT01945775, NCT02326844) [113,114,115]. Importantly, BMN673 as a monotherapy showed a manageable tolerability profile in breast cancer patients who participated in EMBRACA and ABRAZO studies [116,117]. 

Another critical property of BMN673 is the flexible drug treatment time and administration schedule required to achieve strong radiosensitization, at least in certain cell systems. Our results show that a one-hour treatment with BMN673 prior to RT exerted equally strong radiosensitization as continuous treatment during the course of incubation for colony formation, and that even administration several hours after IR radiosensitized cells with impressive efficacy [39]. This property, if proven general, will be very useful in the clinic, as it simplifies the effective combination of BMN673 treatment with radiation exposure in the hectic environment of a radiation therapy facility. Such administration-flexibility has not been reported for other radiosensitizers and requires in-depth analysis of its mechanistic underpinnings.

In addition, BMN673 specifically sensitized cancer cells to RT, whereas normal human cell lines were not sensitized [39], again a favorable parameter in the clinic, promising reduced normal tissue toxicity. The mechanistic underpinnings of this interesting observation require further in-depth investigations as well.

The above studies, in aggregate, highlight the strong potential of BMN673 as a promising radiosensitizer and warrant careful clinical trials to harness such potential in the clinic. To date, 49 clinical trials combining PARPi with RT are registered, which mainly involve testing of veliparib [118,119,120,121,122] or olaparib [123] (Table 1). However, only three of these trials, out of which two are still in the initial phase, involve testing of BMN673 (Table 2). Thus, clinical data supporting the radio-sensitization potential of BMN673 is currently lacking.

The following sections summarize the most prominent mechanisms underpinning the strong radiosensitization potential of BMN673.

## 6. Shift of Balance to Error-Prone DSB Repair Pathways Is Likely the Key to BMN673-Mediated Radiosensitization

Among the multiple lesions induced by IR in the DNA, DSBs are the most deleterious and are evolutionarily hardwired to provoke cellular responses affecting almost every aspect of the cellular metabolism that is known as DDR (see above) [42,124]. To maintain genome integrity, cells of higher eukaryotes utilize four major DSB repair pathways: HR, c-NHEJ, alt-EJ, and SSA, which have different cell cycle and homology dependencies and operate with different fidelity and kinetics [42,125]. By measuring the surrogate DSB marker, γ-H2AX, BMN673 was shown in numerous studies to increase foci load after IR and to inhibit DSB repair [39,40,126]. More specifically, our study revealed that BMN673 shifted the balance of DSB processing from c-NHEJ to error-prone repair pathways, such as alt-EJ, possibly explaining thus the observed radiosensitization [39]. Evidence supporting this postulate is summarized next.

### 6.1. BMN673 Increases DNA End-Resection after IR

DNA end-resection plays a pivotal role in DSB repair pathway choice. DSB repair pathways can be broadly classified as resection-dependent, or resection-independent. During resection an extended overhang of 3′-ssDNA is generated, which serves HR, alt-EJ, and SSA, while strongly antagonizing c-NHEJ [127,128]. Resected DNA ends are promptly coated and stabilized by RPA. In BMN673-treated cells exposed to low, clinically relevant IR doses, a 3- to 4-fold increase in RPA70 foci was observed, suggesting that a larger proportion of the induced DSBs was resected [39]. In this type of studies, it is important to carefully discriminate between increase in RPA foci numbers, which indicates an increase in the proportion of DSBs that undergo resection, and increase in focus intensity, which reflects increased resection at DSBs already shunted to resection dependent processing. Only the former result can be interpreted as a shift in balance from resection-independent to resection-dependent processing. 

Another study also reported increase in RPA foci numbers in irradiated cells treated with BMN673 [129]. Notably, in contrast to BMN673, a study reported that olaparib decreased RPA70 foci formation and promoted c-NHEJ by facilitating DNA-PKcs phosphorylation [130]. This result was confirmed in a more recent study [81]. Thus, enhanced resection is a unique property of BMN673 and is likely to be the key factor of DSB repair pathway balance-disruption observed. It suggests that suppression of c-NHEJ may be associated with enhanced utilization of HR, a topic that is discussed next. 

### 6.2. Key Questions Remain Unanswered Regarding the Effects of BMN673 on HR

HR is the only DSB repair pathway that restores the genome with high fidelity, becoming operational in the S -and G_2_-phases of the cell cycle. We recently reported that the contribution of HR rapidly increases to over 50% at doses between 0.1 and 2.0 Gy, while it becomes undetectable above 10 Gy [131]. For HR to initiate, the Rad51 recombinase, aided by the Rad51 paralogues, must displace RPA from ssDNA to form the Rad51 nucleoprotein filament that invades the sister chromatid [132,133,134]. 

Our results showed that BMN673 treatment caused a marked increase in Rad51 foci, suggesting elevated engagement of HR–as may be expected from the increased resection observed [39]. Similar findings were also reported by Caron M. et al. [129]. These results indicate that the contribution of HR increases in BMN673 treated cells, which contradicts the observed strong radiosensitization [39]. Indeed, increased engagement of an error-free DSB repair pathway is naturally expected to increase rather than decrease the survival of irradiated cells. During HR engagement analysis through RAD51 foci scoring, there are instances of combining outcomes at different endpoints, and the inference is made that increased RAD51 foci actually reflect futile HR, in the sense that it most likely remains incomplete; i.e., at some point the DNA ends are shunted for processing by error-prone repair pathways [135]. More work is urgently needed to quantitate the levels of productive and non-productive HR in BMN673-treated cells. 

Another approach to functionally measure effects of various treatments on HR utilizes reporter cell lines with integrated constructs (e.g., DR-GFP) reporting processing by HR of an I-SceI-induced DSB [136]. Using such assays, it has been reported that olaparib decreases HR [81,137]. The same reporter assay failed in our hands to detect impact on HR after BMN673 treatment [39]. These results appear to contradict results with RAD51 foci formation and suggest peculiarities in the effect of BMN673 on HR that require more work for their elucidation. 

When comparing results obtained by scoring RAD51 foci in irradiated cells with those obtained with reporter cell lines, it must be kept in mind that the latter assay measures effects on repair of an I-SceI-induced “clean” DSB [136]. It is likely that the damaged nucleotides typically present in IR-induced DSBs, generate different processing requirements that are differentially affected by BMN673. This aspect requires further investigations as well. It has been shown that HR preferentially repairs secondary replication induced DSBs after exposure to IR [138]. It would also be interesting to investigate how PARPi and in particular BMN673 modulate induction and repair by HR of such secondary, replication-induced DSBs.

Taken together, these results allow the tentative conclusion that BMN673 leaves HR unchanged at “clean” DSBs, but increases its engagement in IR-induced DSBs. It remains, however possible that in irradiated, BMN673-treated cells, a fraction of HR engagements is unproductive. Either way, the anticipated residual HR activity in BMN673 treated cells predicts benefits in cancer management from a combination of BMN673 with agents inhibiting HR, e.g., ATR inhibitors. 

### 6.3. BMN673 Inhibits c-NHEJ

In mammals, c-NHEJ is the main repair pathway for DSB processing and operates with fast kinetics and efficiency, throughout the cell cycle [42,139,140,141]. Inhibition of c-NHEJ is shown to confer marked radiosensitivity in a wide range of cells [142,143,144]. We recently reported that at low doses of IR, c-NHEJ is partly suppressed by mechanisms that remain to be characterized, giving thus way to HR [131,145,146]. On the other hand, at high IR doses, HR is almost completely suppressed, leaving c-NHEJ as the clearly dominant DSB repair pathway [131,146].

BMN673 was found to inhibit c-NHEJ at high IR doses with efficiency similar to that achieved by DNA-PK inhibitors [39]. It is also highly relevant that BMN673 exerts radiosensitization equivalent to that of c-NHEJ deficient cells [39]. This result, although currently documented only in a few cell lines, is significant, as it is in line with the increase in resection observed, which will naturally suppress c-NHEJ. 

Strikingly, in line with this conclusion, the “end-protector” 53BP1, an antagonist of BRCA1-CtIP-mediated end-resection, is suppressed by BMN673 at low IR doses [39]. This is in line with the observed suppression of c-NHEJ at high IR doses and predicts the increased engagement of BRCA1-CtIP-dependent HR at low IR doses. Since these experiments were carried out in the low-IR-dose region, they suggest that suppression of c-NHEJ occurs in a wide range of doses. Additional studies showed that BMN673 treatment strongly increased end-resection and decreased the accretion of 53BP1 and RIF1 proteins [129]. 

C-NHEJ is mechanistically suppressed at resected DNA ends owing to the low binding efficiency of Ku heterodimer on processed ends [147]. Thus, decreased 53BP1 accretion combined with extensive end-resection underpins the suppression of c-NHEJ by BMN673. It remains however open which factor of c-NHEJ is inhibited, or which resection factor is strengthened by BMN673 to mediate such profound shifts in DSB end-processing upon irradiation. Molecular characterization of this mechanism will not only advance our understanding of the mode of BMN673 action, but will also inform further search for PARPi with similar, or even improved characteristics. It is, however, relevant to point out that when PARPi are administered to HR-deficient cancers, a contribution of HR to DSB repair is a priori ruled out. 

It is worth mentioning here that in contrast to combinations with IR, BMN673 treatment applied as a monotherapy resulted in elevated levels of 53BP1 foci [39,148,149,150], indicating enhanced c-NHEJ activity in this setting. More studies are warranted to elucidate the impact of BMN673 on c-NHEJ, either when used as a monotherapy or when combined with IR.

### 6.4. BMN673 Shunts DSBs to Error-Prone Alt-EJ

Alt-EJ is a collective term used to describe DSB processing that cannot be attributed to HR or c-NHEJ, and which engages as a backup when DSB ends are released owing to failures of these repair pathways [124]. Although alt-EJ may, in principle, also engage as a “first line” DSB repair pathway [151], when this happens, it is important to inquire first why these DSB-ends were “ignored” by mechanisms as efficient as c-NHEJ or HR. Although alt-EJ can, in principle, function on unresected DSB ends [152], it benefits from short-range resection and is therefore dependent on the CtIP/MRN complex. Other proteins implicated in alt-EJ include PARP-1 and Ligases1/3, which are involved in DSB sensing and ligation, respectively [80,153,154]. Polθ is a newcomer to alt-EJ but has raised enormous interest owing to its unusual properties that appear critical to the joining of resected DSB ends that have escaped HR or c-NHEJ. Thus, HR-deficient tumors strongly rely on the function of Polθ, in a PARP-1-dependent manner [155]. Polθ facilitates the annealing of resected 3′-tails, and extends one 3′ DNA-end using the annealing partner as a template to facilitate the final ligation step using the activity of either Lig1 or Lig3 [156]. 

Compared to c-NHEJ, alt-EJ operates with slower kinetics and lower efficiency, and with much greater error-proneness. Consequently, when DSBs are processed by alt-EJ, joining of “wrong”, unrelated DSB ends can occur, drastically increasing thus the probability of chromosomal translocation formation [157]. Indeed, several studies have implicated alt-EJ in chromosomal translocation formation [77,78,158,159,160].

Since PARP-1 plays a vital role in the DSB sensing step of alt-EJ, inhibition of PARP-1 using PARPi generally reduces the frequency of chromosomal translocations. Thus, PARPi such as PJ34, AG14361 and the clinically relevant olaparib, have shown to decrease translocations in irradiated cells [39,77,78,158,159]. By contrast, here again, BMN673 exerts the opposite effect, strongly promoting the formation of translocations after IR [39]. These findings support the notion that cells retreat to alt-EJ, as the ultimate backup, to process DSBs because canonical repair pathways, in this case most likely, predominantly c-NHEJ, failed. Of course, this happens at the very high price of increased incidence of chromosomal translocations. Why, here, BMN673 is again an outlier will require further studies.

Translocations are one of the critical drivers of oncogenesis and the culprits of cell death after exposure of cells to IR [161]. Our study showed that BMN673 treatment increases the frequency of translocations after IR in cancer but not in normal human cells [39], in line with the observed tumor-specific BMN673-mediated radiosensitization. Despite its error-proneness, alt-EJ remains an important mechanism of genome maintenance, because the severity of consequences of the only remaining alternative, a complete lack of DSB processing, are much more severe. Indeed, it appears that cells of higher eukaryotes, are wired to remove all DSBs from their genome using any combination of available DSB repair pathways, provided DNA ends are, or come, in close proximity. Indeed, this design may be aided by the increased chromatin mobility observed after the induction of DSBs [162]. 

A previous study tested alt-EJ using reporter cell lines and concluded that suppressing PARP-1-dependent alt-EJ by olaparib correlates positively to its radiosensitizing effect [33]. This observation, and our results of increased alt-EJ-mediated translocation formation in BMN673-treated cancer cells [39] suggests that BMN673-mediated radiosensitization can be further enhanced by targeting alt-EJ proteins other than PARP-1. It will be intriguing in this regard to examine, combinations of PARPi with readily available Polθ inhibitors that undergo clinical testing [163,164].

## 7. Speculations on the Mechanism of BMN673-Induced Radiosensitization

Retention of inhibited PARP on DNA lesions (the so-called PARP trapping) has been proposed as a key parameter of PAPRi-induced cytotoxicity underpinning the effects of monotherapy applications. PARPi-mediated inhibition of Parp1 auto-PARylation prevents its removal from DNA lesions and leads to replication fork collapse in S-phase cells, or to the suppression of other aspects of the DNA metabolism [95]. BMN63 has been reported as the most potent inducer of PARP-1 trapping, compared to other clinically relevant PARPi, and this property is often causally correlated to its increased cytotoxicity [95]. 

When analyzing the effects of PAPRi, it is important to address separately effects on cytotoxicity that is enhanced in HR deficient cells, and effects on radiosensitivity and DSB processing reviewed here, as they may be caused by different mechanisms. Thus, while it is possible that the enhanced trapping ability of BMN673 explains its toxicity in HR deficient cancers [8,11], it may not underpin, in its non-specific form, the notable radiosensitizing potential. It is relevant in this regard that the PARP trapping ability of clinical PARPi has been mostly demonstrated using DNA alkylating agents such as MMS that are likely to generate lesions strongly inhibiting DNA replication [40,95,165]. Similar forms of PARP1 trapping have not been reported for irradiated cells, and indeed our experiments failed to generate unequivocal evidence for trapping. 

Can Parp1 trapping explain the above observed effects of BMN673 on DSB processing? The shift of DSB repair from c-NHEJ to HR and alt-EJ that requires extensive processing of DNA ends rules out general Parp1 trapping stopping all processing at the DSB end as a mechanism. Indeed, BMN673 mediated excessive resection and accretion of RPA70 and Rad51 after IR [39] is plausible only when DSB ends are not blocked by trapped PARP-1. Therefore, if Parp1 trapping underpins the above discussed DSB repair effects, it must be happening in a highly specific manner centering on c-NHEJ. We hypothesize that after BMN673 treatment, Parp1 inhibits the binding of Ku to DNA ends. Indeed, there is evidence that Parp1 precedes Ku at the DSB end [80], and it is therefore possible that the engagement of Ku is changed by trapped Parp1. Existing evidence suggests that this effect is observed both at high as well as at low doses of IR. We further hypothesize that the interaction of inhibited Parp1 is highly specific for Ku and that it leaves unchanged, or even facilitates all processes associated with DNA end resection. 

It has been shown recently that p97 ATPase/segregase extracts BMN673-induced trapped PARP-1 from chromatin, a process that requires post-translational modification of PARP-1 by PIAS4-mediated SUMOylation and RNF4-mediated ubiquitination [166]. The chromatin remodeler ALC1 (amplified in liver cancer 1) could also remove inactive PARP1 through binding to readily PARylated chromatin prior to PARPi treatment. Moreover, ALC1 is implicated in the recruitment of Rad51, Ku70, RPA and MRE11 to DNA damage sites, thus acting upstream of DSB repair pathway choice after removing trapped PARP [167]. 

The above evidence in aggregate shows that BMN673 indeed traps Parp1 at the DSBs and provides clues on possible mechanisms to undo this trapping. It leaves however open the mechanism of how BMN673 specifically inhibits c-NHEJ and enhances in a directly compensating manner, resection-dependent DSB processing. The elucidation of the mechanisms underpinning these effects should be given high priority. 

Figure 3 summarizes the postulated altered balance in DSB repair utilization after treatment of irradiated cells with BMN673, based on the assumption that it inhibits c-NHEJ. The color zones in the bars show the fraction of DSBs processed by each DSB repair pathway, except SSA that has not been sufficiently analyzed for BMN673 effects. It is drawn on the assumption that only very few DSBs remain unprocessed and that this fraction slightly increases with dose. It is also considering G_2_-phase cells to include effects on HR. 

At low IR doses, c-NHEJ and HR repair similar fractions of DSBs, with alt-EJ having a small contribution [131,145,146]. BMN673 treatment at low IR doses inhibits c-NHEJ and shifts the balance towards HR and to alt-EJ that shows robust increases [39]. At high IR doses, HR is inherently suppressed and the majority of DSBs are processed by c-NHEJ, and to a much lesser extent by alt-EJ [131,146]. BMN673 treatment after exposure to high IR doses also inhibits c-NHEJ, but leaves now the majority of DSBs to be processed by alt-EJ [39]. 

## 8. Summary and Conclusions

In the current review, we discussed the mechanisms underpinning the superior radiosensitizing potential of BMN673, as compared to other clinically relevant PARPi. In addition, we discussed the critical properties of BMN673 that pave the way for testing it in clinical settings as an adjuvant to RT. BMN673 inhibits c-NHEJ activity at low and high IR doses by promoting DSB end resection. As a result, resection-dependent pathways (HR and alt-EJ) show increased activity (Figure 3). This is evident by increased accretion of Rad51 foci and elevated frequency of translocations after irradiation, when cells were additionally treated with BMN673. Under such circumstances, alt-EJ supersedes HR resulting in error-prone repair and robust radiosensitization.

The potential of combining BMN673 with RT is evident from several ongoing pre-clinical studies, but the clinical evidence supporting this approach is scarce. The combination has the potential to generate a paradigm shift and to establish BMN673 as a promising radiosensitizer. Our in vitro results, as well as results from other pre-clinical studies (described in Section 5), make a strong argument in favor of such a combination. Furthermore, combining BMN673 with RT in the clinic may also pave the way to the treatment of HR proficient tumors using PARP inhibition.

## Figures and Tables

**Figure 1 cancers-14-05619-f001:**
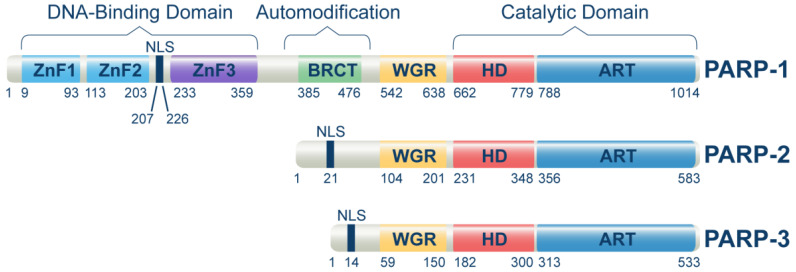
Domain structure of human PARP-1, PARP-2, and PARP-3 proteins. The domain locations within the full-length proteins were extracted from the uniport database by using the following entries for PARP-1 (P09874), PARP-2 (Q9UGN5), and PARP-3 (Q9Y6F1). Main protein domains involved in DNA-binding, auto-modification, and catalytic functions are indicated. ZnFI: zinc finger I domain, ZnFII: zinc finger II domain, ZnFIII: zinc finger III domain, BRCT: BRCA1 C-terminal domain, WGR: Trp-Gly-Arg domain, HD: Helical domain, ART: (ADP-ribosyl) transferase domain. ZnFI and ZnFII help to recognize and bind to DNA damage sites, while ZnFIII activates the enzyme upon DNA binding. The BRCT domain of PARP-1 is required to localize XRCC1 and XRCC1-complexed DNA repair factors to DNA damage sites.

**Figure 2 cancers-14-05619-f002:**
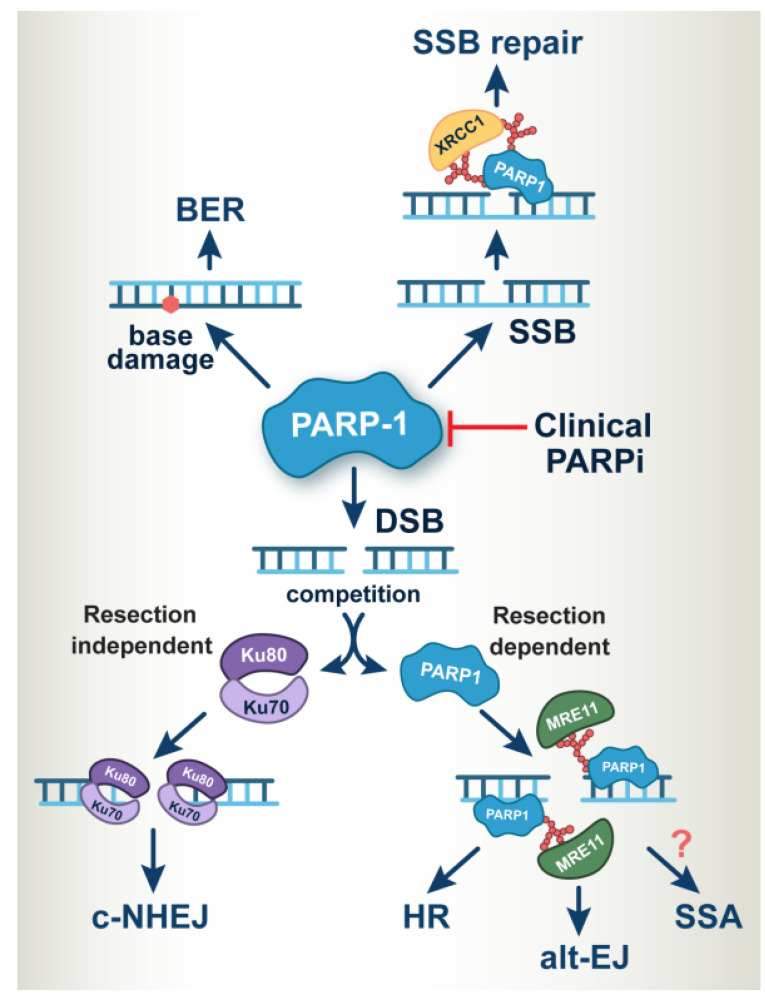
A central and critical role of PARP-1 in DNA repair. PARP-1 is the prime target of clinical PARPi and mediates multiple DNA repair processes (Please refer to Section 2 for details).

**Figure 3 cancers-14-05619-f003:**
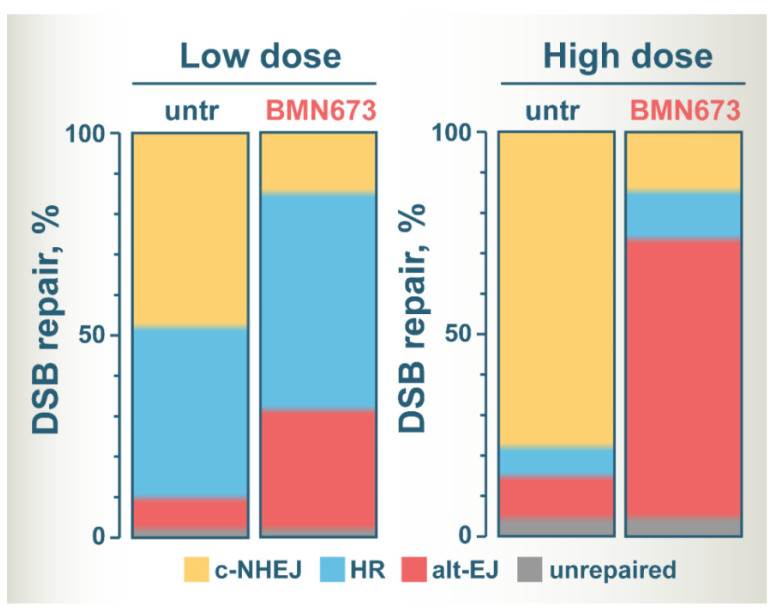
A model of altered DSB repair pathway balance at low and high IR doses in BMN673-treated cells. See text for details.

**Table 1 cancers-14-05619-t001:** Published Clinical Studies on PARPi combined with RT.

Trial Phase	Disease Setting	HR Status	Treatments	Eligible Enrolled Patients	Efficacy	Ref.
I	Inflammatory or loco-regionally recurrent breast cancer	N/A	RT + veliparib	30	3-year OS-56.6%, PFS -50%	[118]
I	Locally advanced rectal cancer	N/A	RT + capecitabine + veliparib	32	Tumor downstaging after surgery-71%; pCR-29%	[119]
II	Brain metastases from NSCLC	N/A	WBRT +/− veliparib	Randomization (1:1:1);	WBRT alone	[120]
				103 in 50mg veliparib arm, 102 in 200 mg veliparib arm, 102 in control arm	Median OS: 185 days	
					WBRT+50 mg veliparib	
					Median OS: 209 days (*p* = 0.933)	
					WBRT+200 mg veliparib	
					Median OS: 209 days (*p* = 0.909)	
I/II	Diffuse pontine gliomas	N/A	RT + veliparib	65	Phase I	[121]
					PR(0%) SD-91.7% PD(8.3%)	
					Phase II	
					PR(13.2%) SD-71.7% PD-9.4%	
II	Unmethyated MGMT glioblastoma	N/A	RT+ temozolomide +/− veliparib	Randomization (2:1)	Without veliparib	[122]
				84 in veliparib arm and 41 in control arm	PFS at 6 months-31%. Median OS: 12.8months	
					With veliparib	
					PFS at 6 months (46%). Median OS: 12.7 months	
I	Head and neck squamous cell carcinoma	N/A	olaparib + RT + cetuximab	15	Median OS: 37 months	[123]
					2-year OS (72%), PFS (63%)	

N/A: not applicable; RT: radiotherapy; OS: overall survival; PFS: progression free survival; pCR: pathological complete response; WBRT: whole brain radiation therapy; PR: partial response; SD: stable disease; PD: progressive disease.

**Table 2 cancers-14-05619-t002:** Registered clinical studies of BMN673 in combination with RT.

Trial Phase	Disease Setting	HR Status	Treatments	Trial Status	Identifier
I	Extensive-stage small cell lung cancer	N/A	BMN673 + low dose consolidative thoracic RT	Recruiting	NCT04170946
I	Locally recurrent gynecologic cancers	N/A	BMN673 + RT	Recruiting	NCT03968406
II	Metastatic triple negative breast cancer	gBRCA 1/2 negative	BMN673 + Atezolizumab + SBRT	Recruiting	NCT04690855

N/A: not applicable; RT: radiotherapy; gBRCA: germline Breast cancer gene 1/2; SBRT: Stereotactic body radiation therapy.

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
