# Peer review of "BMN673 Is a PARP Inhibitor with Unique Radiosensitizing Properties: Mechanisms and Potential in Radiation Therapy"

_cancers, 2022, doi:10.3390/cancers14225619_

Round 1
Reviewer 1 Report
In this current review, the authors discuss nicely the understanding we have on PARP protein inhibitors and especially BMN673 as the most prominent one. The PARP inhibitors target single strand break repair and indirectly may target also base excision or nucleotide excision repair. The review is well thought and organized. The manuscript could benefit from the following additions and responses :
1. The toxicity for normal tissue is always a great problem. Do we have any evidence for normal cells or even better patients that there is minimum toxicity or accepted at least?
2. PARP 1 and 2 are enzymes participating not only in the processing of strand breaks but also oxidative lesions through BER or NER. Are the authors aware of such studies like https://www.ncbi.nlm.nih.gov/pmc/articles/PMC4799059/of studies documenting this with BMN673?
3. The idea of PARPi becomes more important under synthetic lethality. Maybe the authors can discuss discuss this.
4.Just a minor comment, in the abstract the authors need to explain words like PARP, or PARPi to make it easier for the reader.
Author Response
Cancers-1985744
Response to Reviewers' Comments
We are thankful to both reviewers for the overall positive evaluation of our review article. Their comments and criticisms helped us improve the manuscript. Below we provide a point-by-point response to Reviewers’ comments. We submit a “clean” copy of the manuscript and include a “marked copy” generated using the “compare” function of “Word” as “Supplementary Material” to facilitate the visualization of changes made.
Our specific responses are as follows:
Reviewer #1
In this current review, the authors discuss nicely the understanding we have on PARP protein inhibitors and especially BMN673 as the most prominent one. The PARP inhibitors target single strand break repair and indirectly may target also base excision or nucleotide excision repair. The review is well thought and organized. The manuscript could benefit from the following additions and responses:
We thank the reviewer for the positive assessment of our review article.
- The toxicity for normal tissue is always a great problem. Do we have any evidence for normal cells or even better patients that there is minimum toxicity or accepted at least?
This is indeed a very critical aspect of employing BMN673 in the clinic. We would like to attract the reviewer’s attention to the paragraph (lanes 236-247) in which we describe that BMN673 as a monotherapy in clinical studies (EMBRACA and ABRAZO) showed a manageable tolerability profile in breast cancer patients. Results from studies combining BMN673 with radiation are likely to be forthcoming within a year or so.
- PARP 1 and 2 are enzymes participating not only in the processing of strand breaks but also oxidative lesions through BER or NER. Are the authors aware of such studies like https://www.ncbi.nlm.nih.gov/pmc/articles/PMC4799059/of studies documenting this with BMN673?
We thank the reviewer for pointing out this important aspect. PARP-1 and PARP-2 mediated processing of oxidative lesions probably also contributes to the overall responses observed with PARPi in pre-clinical and clinical settings. Therefore, we have included this aspect in section 2 of the revised version (Reference 62).
- The idea of PARPi becomes more important under synthetic lethality. Maybe the authors can discuss this.
We agree with the reviewer that PARPi as monotherapy gained importance in the clinic owing to synthetic lethality with HR defects. Our focus in the current review, is to emphasize the potential of combining PARPi with radiation. In this case we avoid the term of synthetic lethality to describe radiosensitization, because we found it to lead to confusion and misunderstandings. In the revised version, we highlighted the concept of synthetic lethality in monotherapy approaches and added additional references to the introduction section. We use the terms radiosensitization or synergism to describe effects of PAPRi on radiosensitivity.
4.Just a minor comment, in the abstract the authors need to explain words like PARP, or PARPi to make it easier for the reader.
We have explained the terms in the revised version as suggested by the reviewer.
Reviewer 2 Report
Soni et al’s review manuscript is entitled as “BMN673 is a PARP inhibitor with unique radiosensitizing 2 properties: Mechanisms and potential in radiation therapy”. Authors have collected most of the literatures related BMN673 and RT. Considering the importance of BMN673 in radiosensitization of cancer, this review is quite apt and brought out the concept in coherent manner for the readers in this field. This review will surely help the readers in understanding the molecular mechanism of BMN673 in radiosensitization of cancer. The review is well written and quite well structured. However, my major concerns are related to some of the hypothesis, interpretations and projections by the authors (see below). Since genetic defects of PARP and pharmacological inhibitions of PARP by different inhibitors (BMN673. Olaparib, rucaparib and niraparib) show varied response in biological models, the field is still evolving and not correct to conclude/interpret based on few studies (see below). I suggest that authors not to conclude the role of BMN673 in different repair pathways based on few studies. They may subtly hypothesise the mode of action of BMN673 in different DSB repair pathways.
Major and Minor concerns:
1. Authors cited their previous work to emphasize important points that “ (a) treatment with talaozoparib pre-/post-IR conditions also offers radiosensitization and (b) talazoparib BMN673 specifically sensitizes cancer cells but not normal human cell lines to RT” . However, there is a lack of explanations why talazopirb offers sensitization at different time points of RT and why normal cells are not affected? Authors should give some scientific explanations based on the citations or suggest some explanations if no information is available.
2. Line 260-265: It would be interesting and helpful for the readers to see all the clinical trials of PARPi and RT, if these information are given in a table. Clinical trials, cancer types (origin, HR +ve, HR -ve), and citations etc may be given in the table.
- Line 282-292: “In BMN673-treated cells 286 exposed to low, clinically relevant IR doses, a 3- to 4-fold increase in RPA70 foci was observed, suggesting that a larger proportion of the induced DSBs was resected [30]”. Based on their study, authors concluded that enhanced RPA foci in combination treatment may be associated with DSB resections, hence shifting the repair process to HR. However, the possibilities of incurring additional secondary damages induced ssDNAs (replication, PMID: 22505579) during post-IR may also be discussed and included. Just analyzing RPA foci may not distinguish ssDNA at resected DSBs and damaged replication sites, SSBs etc. Besides, enhanced RAD51 foci may also occurs HR-independent manner, as mentioned in line 325, at replication sites. In the absence of extensive literature for mechanistic insight for talazoparib and IR combination treatment, authors should keep the topic open for additional information to fill the gap in the understanding of PARPi-mediated sensitization of cancers to RT.
- Section 6.3. “BMN673 inhibits classical non-homologous end-joining”. Whether BMN inhibits cNHEJ is ambiguous and still debatable. Multiple studies, including authors publication (30, PMID: 31852834, PMID: 33473172, PMID: 35367197) showed that talazoparib itself enhances 53BP1 foci (regarded as NHEJ sites), whereas talazoparib reduces but does not completely abrogate 53BP1 foci in combination with RT. In this regard, reference 113 showed the differential impact of BMN on NHEJ (Ku recruitment, 53Bp1) and HR. However, the later study used very high dose of BMN (>1 mM), which may have pleiotropic effects on other factors. This whole section was based on two references (30, 113), so authors may summarise these two studies for cNHEJ inhibition by BMN673 but more studies are warranted to validate the precise role of BMN/talazoparib in cNHEJ. The title of this section may not be correct or generalized based on the available studies (30, PMID: 31852834, PMID: 33473172, PMID: 35367197), - all these studies suggest that talazoparib alone treatment enhances cNHEJ. Author should include other views (PMID: 31852834, PMID: 33473172, PMID: 35367197), and conclude IR-related studies with the scope of including additional twist in this field in the future.
- Since genetic defects in PARP1 and pharmacological effects of different PARPi (BMN, Olaparib, PJ34, AG14361 etc) have varied cellular and molecular phenotypes, authors should be careful in some of the interpretations made in the review. For example, “……It leaves however completely open the mechanism as to how BMN673 specifically inhibits c-NHEJ and enhances in direct balancing response resection-dependent processing. The elucidation of the underpinning mechanisms should be given high priority.” As mentioned above, it is hard to interpret that BMN inhibits cNHEJ, in the absence of solid evidence. In fact, a single treatment of BMN enhances 53BP1/Ku foci in different cancer cells.
- “………This observation and our results of increased alt-EJ-mediated translocation formation in BMN673-treated cancer cells suggest that BMN673-mediated radiosensitization can be further enhanced by targeting alt-EJ proteins other than PARP-1.” Citation for authors' results is required here.
- Figure 3 is purely speculative? Whether experimental evidence available to support the level of quantifications (Y-axis) for different repair pathways? In my opinion, only a hypothetical model should be projected for BMN673 in Fig.3, rather than what is shown in Fig.3 in the MS.
- Since only very few literature is available for BMN + RT, it is prudent to dedicate a small section on summarizing some of the interesting literature related to BMN673 alone treatment in cancers and show the mechanism of BMN673 action for cNHEJ (PMID: 31852834, PMID: 33473172, PMID: 35367197), alt-EJ and HRR, prior to section 6. Further the similarity and differences in the action of BMN673 alone and combination treatment with RT should be focussed on in Section 6.
- Line 343: “We recently reported…….”. Citation is required for this statement.
- Line 102: “PARY-lation” should be “PARylation”
- Line 249: “at lease” should be “at least”
Author Response
Cancers-1985744
Response to Reviewers' Comments
We are thankful to both reviewers for the overall positive evaluation of our review article. Their comments and criticisms helped us improve the manuscript. Below we provide a point-by-point response to Reviewers’ comments. We submit a “clean” copy of the manuscript and include a “marked copy” generated using the “compare” function of “Word” as “Supplementary Material” to facilitate the visualization of changes made.
Our specific responses are as follows:
Reviewer #2
Soni et al’s review manuscript is entitled as “BMN673 is a PARP inhibitor with unique radiosensitizing properties: Mechanisms and potential in radiation therapy”. Authors have collected most of the literatures related BMN673 and RT. Considering the importance of BMN673 in radiosensitization of cancer, this review is quite apt and brought out the concept in coherent manner for the readers in this field. This review will surely help the readers in understanding the molecular mechanism of BMN673 in radiosensitization of cancer. The review is well written and quite well structured. However, my major concerns are related to some of the hypothesis, interpretations and projections by the authors (see below). Since genetic defects of PARP and pharmacological inhibitions of PARP by different inhibitors (BMN673. Olaparib, rucaparib and niraparib) show varied response in biological models, the field is still evolving and not correct to conclude/interpret based on few studies (see below). I suggest that authors not to conclude the role of BMN673 in different repair pathways based on few studies. They may subtly hypothesise the mode of action of BMN673 in different DSB repair pathways.
We thank the reviewer for the overall positive assessment of our review article. Below we provide point-by-point responses to the issues raised by the reviewer and the steps we took to alleviate the above formulated concerns.
Major and Minor concerns:
1. Authors cited their previous work to emphasize important points that “ (a) treatment with talazoparib pre-/post-IR conditions also offers radiosensitization and (b) talazoparib BMN673 specifically sensitizes cancer cells but not normal human cell lines to RT” . However, there is a lack of explanations why talazoparib offers sensitization at different time points of RT and why normal cells are not affected? Authors should give some scientific explanations based on the citations or suggest some explanations if no information is available.
We have extended these passages in the paper and offer explanations, when possible. Frequently, we also point out that more work will be needed to conclusively elucidate the underpinning mechanisms.
2. Line 260-265: It would be interesting and helpful for the readers to see all the clinical trials of PARPi and RT, if these informations are given in a table. Clinical trials, cancer types (origin, HR +ve, HR -ve), and citations etc may be given in the table.
This is an excellent suggestion. We have included 2 Tables in the revised version of our paper. Table 1 summarizes published clinical trials combining PARPi and RT. Table 2 summarizes the registered clinical studies of BMN673 in combination with RT.
3. Line 282-292: “In BMN673-treated cells exposed to low, clinically relevant IR doses, a 3- to 4-fold increase in RPA70 foci was observed, suggesting that a larger proportion of the induced DSBs was resected [30]”. Based on their study, authors concluded that enhanced RPA foci in combination treatment may be associated with DSB resections, hence shifting the repair process to HR. However, the possibilities of incurring additional secondary damages induced ssDNAs (replication, PMID: 22505579) during post-IR may also be discussed and included. Just analyzing RPA foci may not distinguish ssDNA at resected DSBs and damaged replication sites, SSBs etc. Besides, enhanced RAD51 foci may also occurs HR-independent manner, as mentioned in line 325, at replication sites. In the absence of extensive literature for mechanistic insight for talazoparib and IR combination treatment, authors should keep the topic open for additional information to fill the gap in the understanding of PARPi-mediated sensitization of cancers to RT.
This is a very relevant point and we edited different places in the revised manuscript to make clear to the reader that the topic is still in progress and more work will be needed to address it from various angles. In section 6.2 of the revised paper we discuss the specific topic raised by the Reviewer.
4. Section 6.3. “BMN673 inhibits classical non-homologous end-joining”. Whether BMN inhibits cNHEJ is ambiguous and still debatable. Multiple studies, including authors publication (30, PMID: 31852834, PMID: 33473172, PMID: 35367197) showed that talazoparib itself enhances 53BP1 foci (regarded as NHEJ sites), whereas talazoparib reduces but does not completely abrogate 53BP1 foci in combination with RT. In this regard, reference 113 showed the differential impact of BMN on NHEJ (Ku recruitment, 53Bp1) and HR. However, the later study used very high dose of BMN (>1 mM), which may have pleiotropic effects on other factors. This whole section was based on two references (30, 113), so authors may summarise these two studies for cNHEJ inhibition by BMN673 but more studies are warranted to validate the precise role of BMN/talazoparib in cNHEJ. The title of this section may not be correct or generalized based on the available studies (30, PMID: 31852834, PMID: 33473172, PMID: 35367197), - all these studies suggest that talazoparib alone treatment enhances cNHEJ. Author should include other views (PMID: 31852834, PMID: 33473172, PMID: 35367197), and conclude IR-related studies with the scope of including additional twist in this field in the future.
We thank the Reviewer for this insightful comment in such an important topic. We have included and discussed the studies pointed out by the reviewer in section 6.3 of the revised paper. We have also modified the title of this section based on the fact that BMN673 inhibits c-NHEJ, specifically after IR. We conclude that more work is need on this topic.
5. Since genetic defects in PARP1 and pharmacological effects of different PARPi (BMN, Olaparib, PJ34, AG14361 etc) have varied cellular and molecular phenotypes, authors should be careful in some of the interpretations made in the review. For example, “……It leaves however completely open the mechanism as to how BMN673 specifically inhibits c-NHEJ and enhances in direct balancing response resection-dependent processing. The elucidation of the underpinning mechanisms should be given high priority.” As mentioned above, it is hard to interpret that BMN inhibits cNHEJ, in the absence of solid evidence. In fact, a single treatment of BMN enhances 53BP1/Ku foci in different cancer cells.
This comment directly follows from comment 4 and the revised passage refers to the information included in the response to comment 4. We have rephrased several passages to indicate the hypothetical nature of the model.
6. “………This observation and our results of increased alt-EJ-mediated translocation formation in BMN673-treated cancer cells suggest that BMN673-mediated radiosensitization can be further enhanced by targeting alt-EJ proteins other than PARP-1.” Citation for authors' results is required here.
Relevant citation is included in the revised version of the paper.
7. Figure 3 is purely speculative? Whether experimental evidence available to support the level of quantifications (Y-axis) for different repair pathways? In my opinion, only a hypothetical model should be projected for BMN673 in Fig.3, rather than what is shown in Fig.3 in the MS.
Figure 3 summarizes limited experimental evidence gained using a few cell lines – as the Reviewer pointed out in her/his introductory statement. We now indicate these facts in the revised manuscript and include references to the work that helped us formulate it.
8. Since only very few literature is available for BMN + RT, it is prudent to dedicate a small section on summarizing some of the interesting literature related to BMN673 alone treatment in cancers and show the mechanism of BMN673 action for cNHEJ (PMID: 31852834, PMID: 33473172, PMID: 35367197), alt-EJ and HRR, prior to section 6. Further the similarity and differences in the action of BMN673 alone and combination treatment with RT should be focussed on in Section 6.
We thank the Reviewer for the suggestion. We considered the studies pointed out by the reviewer (PMID: 31852834, PMID: 33473172, PMID: 35367197) in section 6.3 and put them in context to the results with combined BMN673 and RT treatment.
9. Line 343: “We recently reported…….”. Citation is required for this statement.
The relevant citations are incorporated in the revised version.
10. Line 102: “PARY-lation” should be “PARylation”
Corrected.
11. Line 249: “at lease” should be “at least”
Corrected.
Round 2
Reviewer 2 Report
The authors have addressed most of the queries. The manuscript is substantially improved after the revision.
Author Response
We sincerely thank the reviewer for the positive assessment of our review article.